# Affinity Tag Coating Enables Reliable Detection of Antigen-Specific B Cells in Immunospot Assays

**DOI:** 10.3390/cells10081843

**Published:** 2021-07-21

**Authors:** Sebastian Köppert, Carla Wolf, Noémi Becza, Giuseppe A. Sautto, Fridolin Franke, Stefanie Kuerten, Ted M. Ross, Paul V. Lehmann, Greg A. Kirchenbaum

**Affiliations:** 1Research & Development Department, Cellular Technology Limited, Shaker Heights, OH 44122, USA; sebastian.koeppert@fau.de (S.K.); carla.wolf@fau.de (C.W.); noemi.becza@immunospot.com (N.B.); fridolin.franke@immunospot.com (F.F.); paul.lehmann@immunospot.com (P.V.L.); 2Institute of Anatomy and Cell Biology, Friedrich-Alexander University Erlangen-Nürnberg, 91054 Erlangen, Germany; stefanie.kuerten@uni-bonn.de; 3Center for Vaccines and Immunology, University of Georgia, Athens, GA 30602, USA; gasautto@uga.edu (G.A.S.); tedross@uga.edu (T.M.R.); 4Institute of Neuroanatomy, Medical Faculty, University of Bonn, 53115 Bonn, Germany; 5Department of Infectious Diseases, University of Georgia, Athens, GA 30602, USA

**Keywords:** antibody-secreting cell, fluorospot, memory B cell, plasmablast, antigen coating, influenza, EBV, HCMV, SARS-CoV-2, immune monitoring

## Abstract

Assessment of humoral immunity to SARS-CoV-2 and other infectious agents is typically restricted to detecting antigen-specific antibodies in the serum. Rarely does immune monitoring entail assessment of the memory B-cell compartment itself, although it is these cells that engage in secondary antibody responses capable of mediating immune protection when pre-existing antibodies fail to prevent re-infection. There are few techniques that are capable of detecting rare antigen-specific B cells while also providing information regarding their relative abundance, class/subclass usage and functional affinity. In theory, the ELISPOT/FluoroSpot (collectively ImmunoSpot) assay platform is ideally suited for antigen-specific B-cell assessments since it provides this information at single-cell resolution for individual antibody-secreting cells (ASC). Here, we tested the hypothesis that antigen-coating efficiency could be universally improved across a diverse set of viral antigens if the standard direct (non-specific, low affinity) antigen absorption to the membrane was substituted by high-affinity capture. Specifically, we report an enhancement in assay sensitivity and a reduction in required protein concentrations through the capture of recombinant proteins via their encoded hexahistidine (6XHis) affinity tag. Affinity tag antigen coating enabled detection of SARS-CoV-2 Spike receptor binding domain (RBD)-reactive ASC, and also significantly improved assay performance using additional control antigens. Collectively, establishment of a universal antigen-coating approach streamlines characterization of the memory B-cell compartment after SARS-CoV-2 infection or COVID-19 vaccinations, and facilitates high-throughput immune-monitoring efforts of large donor cohorts in general.

## 1. Introduction

A successful and durable adaptive immune response comprises both expansion of short-lived effectors and development of long-lived memory cells. Focusing on the humoral component, this entails differentiation of antigen-specific B cells into antibody-secreting cells (ASC), capable of producing large quantities of affinity-matured immunoglobulin/antibodies and expansion of antigen-experienced, class-switched memory B cells poised to rapidly respond upon future antigen encounter. Thus, to truly appreciate the scope of a given B-cell response, both antigen-specific ASC and the memory cells themselves merit detailed study.

Traditionally, monitoring of B-cell responses has largely relied upon serum antibody measurements due to the ease of sample acquisition and high concentration of antibody in this biological fluid. However, secreted antibody exhibits a relatively short half-life (~3 weeks) in vivo [1,2] and maintenance of serum antibody levels therefore requires continuous replenishment. With the exception of following acute antigen exposure, in which elevated frequencies of short-lived ASC can be found in circulation and transiently alter serum antibody reactivity [3,4,5], long-lived plasma cells (LLPC) residing in the bone marrow and secondary lymphoid tissues are chiefly responsible under steady state for the composition and specificity of serum antibody (reviewed by [6]). LLPC are thought to originate from terminally differentiated B cells that participated in T-cell-dependent germinal center reactions [7,8]. Short-lived ASC are also generated during T-cell-dependent immune responses, yet only a subset of these ASC acquire the appropriate gene expression program and successfully occupy the specialized niche required for long-term survival and maintenance of sustained antibody production (reviewed by [9,10,11]).

Such LLPC and their secreted antibodies represent the first line of humoral defense against re-infection with pathogens such as seasonal influenza [12], Ebola [13], Dengue [14] and most recently the SARS-CoV-2 virus [15]. Moreover, pre-existing antibodies also serve to counter reactivation and subsequent dissemination of latent viral infections such as Epstein–Barr virus (EBV) [16] and human cytomegalovirus (HCMV) [17,18]. In this context, elevated levels of serum antibody reactivity serve as a reliable indicator of antigenic exposure. However, both the size and longevity of an antigen-specific LLPC compartment, along with the magnitude and maintenance of serum antibody titers, are highly variable and also may be pathogen dependent (reviewed by [9,19]). Consequently, while some individuals are endowed with LLPC that maintain stable titers of antigen-specific Ig for decades, in other instances antibody titers eventually wane to below the limit required for protection [20,21], as also appears to be the case after SARS-CoV-2 infection [22,23,24].

In such instances, when pre-existing antibody abundance declines to below protective levels, long-lived memory B cells generated during the primary immune response can serve as the second line of humoral defense (reviewed by [25,26]). Poised to rapidly respond upon antigen re-encounter and undergo robust proliferation, owing to their increased precursory frequency and expression of class-switched, affinity-matured B-cell receptors (BCR), the resulting progeny include both additional long-lived memory B cells that await future antigen encounters and ASC capable of acutely raising antibody titers to combat and limit dissemination of the offending pathogen (reviewed by [27]). Hence, immune monitoring of underlying memory B-cell frequencies, and their endowed antigen-specificity, can provide invaluable insights into whether these cells are capable of conferring immunological protection upon recall.

The study of antigen-specific memory B cells is complicated by their low precursory frequency, and because in the absence of recent antigen encounter these cells exist in a resting state and are indistinguishable from other resting B cells on the basis of surface phenotype (reviewed by [28]). Consequently, techniques that yield single-cell resolution and which leverage the specificity of the BCR expressed by individual B cells are best suited for this task (reviewed by [29]). A commonly utilized approach for the identification of human memory B cells relies on flow cytometry and is dependent on BCR-mediated capture of fluorescently-conjugated antigen probes. While technically complex, this strategy has been implemented successfully for the study of memory B-cell reactivity against targets such as influenza hemagglutinin [30], HIV envelope [31] and most recently the SARS-CoV-2 spike and nucleocapsid proteins [24,32,33]. Moreover, this methodology is conducive to multiplexing since several antigenic probes can be used in parallel with minimal interference [34]. In concert with advancements in the fields of flow cytometry, single-cell sequencing and bioinformatics, antigen probe-based memory B-cell assessments offer an unprecedented window of detail into an individual’s memory B-cell composition (reviewed by [35]). Nevertheless, these data exist at the level of nucleotides and our present ability to translate these paired *IgH*/*IgL* sequences into predicted antigen binding in silico is still limited [36]. Furthermore, despite optimizations improving the scalability of molecular cloning and expression of recombinant monoclonal antibodies (mAbs) obtained through antigen probe-based sequencing efforts, the characterization of individual mAbs isolated from multiple donors is costly, labor intensive and not sufficiently throughput for its practical application in any sizeable immune-monitoring effort.

As noted previously, memory B cells exist in a quiescent state in the absence of recent antigen encounter and importantly do not secrete their individual BCR as soluble antibodies. To overcome this obstacle, several in vitro stimulation protocols have been described that facilitate efficient antigen-independent differentiation of resting memory B cells into ASC [37,38,39,40,41,42,43]. In doing so, these in vitro stimulation protocols provide an alternative strategy for measurement of pre-existing memory B cells on the basis of their secreted antibody reactivity. First described by Czerkinsky [44], the traditional enzyme-linked immunospot (ELISPOT) assay technique enabled enumeration of individual ASC through the capture of their secreted Ig in close proximity to the secreting cell, and culminates with deposition of a precipitating substrate to reveal the individual antibody secretory footprint. Owing to its plate-based format, relatively straight-forward procedural methodology and software-assisted counting algorithms, the ELISPOT technique is ideally suited for large-scale assessments of ASC [45,46]. In agreement, the B-cell ELISPOT technique has successfully been implemented for the measurement of antigen-specific ASC against an array of both foreign [3,4,47,48,49,50,51,52] and self-antigens [53,54,55], and is a common approach for assessing B-cell responses in the context of seasonal influenza vaccination where the complexities of pre-existing serum antibody reactivity exist [56,57]. Subsequent adaptations of the ELISPOT method and usage of detection reagents coupled to fluorochromes with unique excitation and emission spectra are referred to as FluoroSpot. Assessment of B-cell responses is particularly well-suited for the FluoroSpot platform because multiple Ig classes or IgG subclasses can be independently measured in parallel with no ambiguity or interference, through usage of reagents conjugated with distinct fluorophores [58,59].

Using either an ELISPOT or FluoroSpot assay approach (collectively ImmunoSpot), or if a single or multiple antibody class/subclass are measured, the greatest obstacle to detection and subsequent successful enumeration of antigen-specific ASC is achieving a sufficiently high density of antigen coating on the membrane of the assay well itself. Unlike in a B-cell ImmunoSpot assay measuring total ASC, in which the secreted antibody is captured with fixed affinity through capture by polyclonal reagents recognizing multiple epitopes in the constant region of the kappa and lambda light chains, respectively, in an antigen-specific application, the efficiency of antibody/antigen binding and its ultimate retention in close proximity to the secreting cells is intrinsically tied to an individual ASC’s fine specificity. In this regard, inherent differences in ASC affinity for the nominal antigen could account for diversity in ImmunoSpot morphologies [60]. Supporting this notion, and in agreement with the polyclonal nature and diversity of BCR affinities present in a given B-cell repertoire, a spectrum of antigen-specific spot sizes are commonly observed in ImmunoSpot assays [37,61,62,63]. By extension, the capacity to detect antigen-specific ASC spanning a broad spectrum of affinity, along with the sensitivity to reliably detect ASC with reduced functional affinity, using either the ELISPOT or FluoroSpot assay platform, is intimately associated with the antigen-coating component since it is responsible for capture and retention of secreted Ig in close proximity to the secreting cell.

Microplates with polyvinylidene difluoride (PVDF) membranes are considered the gold standard for usage in ImmunoSpot assays due to their porosity and ability to bind high concentrations of antibodies [64,65]. However, PVDF membranes are intrinsically hydrophobic and absorption of biomolecules is mediated primarily through hydrophobic and dipole interactions. Consequently, establishment of antigen-specific B-cell ImmunoSpot assays is often limited by the unique biochemistry of the coating antigen itself; that is, whether it can be efficiently absorbed to the membrane. While increasing the concentration of some coating antigens can improve the quantity of protein absorbed to the membrane, this may be prohibitive due to the associated cost or abundance of antigen required. Lastly, due to their hydrophilic nature, there are antigens that cannot be effectively coated to PVDF membranes even at excessively high concentrations.

In this communication, we describe a universal antigen-coating approach for efficient absorption of recombinantly expressed antigens representing a diverse set of viral pathogens through exploiting their genetically encoded hexahistidine (6XHis) affinity tags. Using murine B-cell hybridomas and primary human memory B cells secreting antibody specificities spanning a spectrum of affinity as model ASC, we highlight improvements in the sensitivity of antigen-specific B-cell FluoroSpot assays, especially in the context of ASC with reduced functional affinity. When combined with the beneficial attributes of the FluoroSpot detection platform, the optimizations in antigen coating reported herein open the door to larger-scale immune-monitoring efforts of underlying memory B-cell reactivity using diverse panels of antigens, along with more sophisticated image analysis approaches that appreciate the attributes of individual ASC.

## 2. Materials and Methods

### 2.1. Human Subjects

Peripheral blood mononuclear cells (PBMC) of human subjects tested in this study were obtained from healthy adults and originate from the ePBMC^®^ library (Cellular Technology Limited (CTL), Shaker Heights, OH, USA). These subjects were recruited by Hemacare (Van Nuys, CA, USA) or American Red Cross (Atlanta, GA, USA) with IRB approvals and were sold to CTL, identifying donors by code only while concealing the subjects’ identities. The PBMC were then cryopreserved and stored in liquid nitrogen until testing.

### 2.2. Polyclonal Human B-Cell Stimulation

Thawing, washing and counting of PBMC was performed according to previously described protocols [66,67] using CTL’s Live/Dead cell-counting suite on an ImmunoSpot^®^ S6 Ultimate Analyzer (CTL). Subsequently, cells were resuspended at 2 × 10^6^ cells/mL in complete B-cell medium (BCM) containing RPMI 1640 (Lonza, Walkersville, MD) supplemented with 10% fetal bovine serum (Gemini Bioproducts, West Sacramento, CA, USA), 100 U/mL penicillin, 100 μg/mL streptomycin, 2 mM l-glutamine, 1 mM sodium pyruvate, 8 mM HEPES (all obtained from Life Technologies, Grand Island, NY, USA) and 50 μM beta-mercaptoethanol (Sigma-Aldrich, St. Louis, MO, USA) containing B-Poly-S™ reagent (TLR7/8 agonist R848 + recombinant human IL-2) (from CTL). Cells were then transferred into tissue culture flasks (Corning, New York, NY, USA, Sigma-Aldrich) and incubated at 37 °C, 5% CO_2_ for five days prior to usage in B-cell ImmunoSpot^®^ assays. Following polyclonal stimulation, cells were washed with PBS and then resuspended in complete BCM at 1–3 × 10^6^ live cells/mL (antigen-specific) or 3 × 10^5^ live cells/mL (total) and used immediately in ImmunoSpot^®^ assays.

### 2.3. Recombinant Proteins

Recombinant hemagglutinin (rHA) proteins encoding A/California/04/2009 (CA/09, H1N1) or A/Texas/50/2012 (TX/12, H3N2) vaccine strains were acquired from the Center for Vaccines and Immunology (CVI) (University of Georgia, Athens, GA, USA) and were described previously [68]. Additionally, full-length SARS-CoV-2 Spike protein [69], along with a truncated version representing the receptor-binding domain (RBD) [70], was also generously provided by CVI at UGA. Recombinant Epstein–Barr virus (EBV) nuclear antigen 1 (EBNA1) protein was purchased from Serion (Würzburg, Germany). Cytomegalovirus gH pentamer complex was purchased from The Native Antigen Company (Oxford, UK). SARS-CoV-2 Spike (S1 fragment) was purchased from Creative Diagnostics (Shirley, NY, USA). Importantly, all recombinant proteins used in this study possessed a genetically encoded hexahistidine (6XHis) affinity tag.

### 2.4. Total (IgA/IgG/IgM) Human B-Cell ImmunoSpot^®^ Assays

For enumeration of antibody-secreting cells (ASC), irrespective of their antigen specificity, cell suspensions were serially diluted 2-fold in duplicates, starting at 3 × 10^4^ live cells/well, in round-bottom 96-well tissue culture plates (Corning, Sigma-Aldrich) and subsequently transferred into assay plates precoated with anti-Igκ/λ capture antibody contained in the human IgA/IgG/IgM Three-Color ImmunoSpot^®^ kit (from CTL). Plates were incubated for 16 h at 37 °C, 5% CO_2_, and plate-bound Ig spot-forming units (SFU), each representing a secretory footprint of an individual ASC, were revealed using the IgA-, IgG- and IgM-specific detection reagents contained in the kit, which was used according to the manufacturer’s instructions.

### 2.5. Antigen-Specific Human B-Cell ImmunoSpot^®^ Assays

For enumeration of antigen-specific ASC, 6XHis-tagged protein antigens were coated directly into 70% (*v*/*v*) EtOH pre-conditioned assay wells at 10 μg/mL in PBS overnight at 4 °C. Alternatively, 6XHis-tagged protein solutions at 10 µg/mL in PBS (unless otherwise specified) were applied to EtOH pre-conditioned wells precoated overnight at 4 °C with 10 µg/mL (unless otherwise specified) anti-His tag antibody (Biolegend, San Diego, CA, USA) and incubated overnight at 4 °C to improve antigen absorption. Following one wash with 150 μL PBS, plates were blocked with 150 μL complete BCM for 1 h at room temperature prior to addition of polyclonally-stimulated PBMC (1–3 × 10^5^ cells/well). Plates were then incubated for 16 h at 37 °C, 5% CO_2_, and SFU were visualized using the human IgA/IgG/IgM Three-Color ImmunoSpot^®^ kit (from CTL) according to the manufacturer’s instructions.

### 2.6. Murine B-Cell Hybridomas

Murine B-cell hybridoma lines secreting (IgG1, κ) monoclonal antibody (mAb) with reactivity against the hemagglutinin (HA) protein of the pandemic H1N1 (pH1N1) A/California/04/2009 (CA/09) influenza vaccine strain were reported previously [71]. These B-cell hybridoma lines were additionally single-cell subcloned upon receipt, and the selected subclone is denoted. Murine B-cell hybridoma lines secreting mAb with reactivity against the Spike protein of SARS-CoV-2 were generated from DBA/2J mice (Jackson Laboratory, Bar Harbor, ME, USA) that were immunized intraperitoneally with heat-inactivated SARS-CoV-2 virus antigen [72] (~100 µL of virus antigen/mouse) adjuvanted with alum hydroxide on day 0, day 21 and day 42. To focus antibody reactivity towards the Spike protein, mice were boosted intraperitoneally with 15 μg of SARS-CoV-2 Spike protein [69] adjuvanted with aluminum hydroxide on day 63. Mice received a final intraperitoneal booster immunization with 15 µg of full-length SARS-CoV-2 Spike protein in PBS on day 77 and fusion of immune splenocytes with SP2/0 myeloma cells was performed four days later, following similar methods as described previously [71,73]. B-cell hybridomas were then single-cell cloned using FACS, as previously described [71]. All B-cell hybridoma lines were cultured in complete BCM at 37 °C, 5% CO_2_.

### 2.7. Murine B-Cell ImmunoSpot^®^ Assays

Murine B-cell hybridoma cells were washed with complete BCM to remove residual mAb in the solution prior to plating into ImmunoSpot^®^ assays. For enumeration of total ASC, irrespective of their antigen specificity, ~100 B-cell hybridomas were seeded into wells precoated with anti-Igκ capture antibody contained in mouse B-cell ImmunoSpot^®^ kits (from CTL). For enumeration of antigen-specific ASC, and comparison of FluoroSpot formation, murine B-cell hybridomas were seeded into wells coated directly with CA/09 rHA or SARS-CoV-2 Spike protein at 10 or 25 µg/mL. Alternatively, CA/09 rHA or Spike proteins at 10 µg/mL were applied to wells precoated with anti-His mAb (IgG2b, κ) (Thermo Fisher, Waltham, MA, USA) as detailed above. Following input of B-cell hybridomas into assay wells, plates were incubated for 16 h at 37 °C, 5% CO_2_, and SFU, each representing a secretory footprint of an individual ASC, were visualized using the IgG1 subclass-specific detection component of the mouse IgG1/IgG2a/IgG2b Three-Color ImmunoSpot^®^ kit (from CTL) according to the manufacturer’s instructions.

### 2.8. ImmunoSpot^®^ Image Acquisition and Counting

Following completion of B-cell ImmunoSpot^®^ assay detection systems, plates were air-dried prior to scanning on an ImmunoSpot^®^ S6 Ultimate Analyzer. SFU were then enumerated using the Basic Count mode of CTL ImmunoSpot SC Studio (Version 1.6.2). As ImmunoSpot^®^ Multi-Color B-cell kits, analyzers and software proprietary to CTL were used in this study; we refer to this collective methodology as ImmunoSpot^®^.

### 2.9. Bivariate Visualization of FluoroSpots

Counted FluoroSpots from replicate wells of an individual B-cell hybridoma line or donor and originating from the same coating condition (direct or affinity capture) were merged into flow cytometry standard (FCS) files using CTL ImmunoSpot SC Studio software, and were subsequently visualized using Flowjo™ (Version 10.6.2) (Ashland, OR, USA).

### 2.10. Statistical Methods

An analysis of variation (ANOVA) with Sidak’s post hoc test was used to identify differences in SFU/well between antigen-coating conditions. (GraphPad Prism 9, San Diego, CA, USA)

## 3. Results

### 3.1. Affinity Tag Capture of SARS-CoV-2 Spike Proteins Improves Antigen-Specific FluoroSpot Assays

We set out to develop a SARS-CoV-2 Spike-specific B-cell ImmunoSpot assay to determine whether this assay format was suitable for immune monitoring of antigen-specific memory B cells following resolution of COVID-19. To monitor B-cell reactivity against the SARS-CoV-2 Spike protein, we evaluated two candidate antigens representing the S1 component or the receptor-binding domain (RBD). Both antigen targets were used previously for evaluating prior infection with SARS-CoV-2 on the basis of serum antibody reactivity [74,75]. Additionally, serum antibody reactivity against the RBD protein is a strong predictor of neutralizing activity [75,76]. As shown in Figure 1A, direct capture of the RBD protein at 10 μg/mL to the assay membrane failed to yield bright and well-defined IgG^+^ FluoroSpots following overnight incubation of pre-stimulated PBMC from four recovered COVID-19 donors. By contrast, IgG^+^ FluoroSpots generated in wells coated in parallel with the same RBD protein solution (at 10 μg/mL) using the affinity capture approach exploiting the genetically encoded hexahistidine (6XHis) tag were readily apparent. Affinity capture of RBD protein not only yielded an increased number of antigen-specific IgG^+^ FluoroSpots, but also led to an enhancement of their intensity and reduction in size (collectively referred to as morphology) relative to FluoroSpots detected in wells in which RBD protein was directly captured. Importantly, IgG^+^ FluoroSpots were not present in wells coated in parallel with the anti-6XHis capture mAb and 6XHis peptide (data not shown).

While all three improvements in B-cell FluoroSpot assay performance and detection of individual ASC are readily perceived visually (increase in spot numbers, increase in spot intensity and improved size definition), they can also be graphically represented in bivariate plots using the flow cytometry standard (FCS) output of ImmunoSpot^®^ data. In addition to increased SFU counts in the affinity capture compared to direct RBD-coated wells (shown in red and black, respectively, in the upper right corner of each panel), an increase in FluoroSpot intensity and reduction in their size was also apparent by the shift to the right and down of the (red) spot clouds for each donor (Figure 1B). In this representation of the data, bright and pristine FluoroSpots will cluster towards the lower right quadrant of intensity vs. size plots. In contrast, large and faint FluoroSpots that are barely countable will locate to the upper left quadrant of such bivariate plots. Importantly, as nearly all RBD-specific FluoroSpots detected in directly coated wells resided in the upper left quadrant, the affinity capture approach enabled unambiguous detection of memory B cells that were barely discernable using a standard antigen-coating approach.

Additionally, we also evaluated these same COVID-19-recovered donors for evidence of antigen-specific B-cell memory targeting the S1 component of the SARS-CoV-2 Spike protein. Whereas direct capture of S1 protein yielded detectable IgG^+^ FluoroSpots and evidence of immunological memory, these FluoroSpots were faint and diffuse (Figure 2A). In comparison, FluoroSpots generated in wells in which the S1 protein was affinity captured via its 6XHis tag appeared more intense and well defined. Consistent with the RBD data, we also detected a greater abundance of S1-specific IgG^+^ FluoroSpots in wells coated through affinity capture and these FluoroSpots exhibited an enhanced morphology (Figure 2B).

Lastly, to more precisely define the improvement realized through testing primary human B cells in affinity-captured versus directly antigen-coated assay wells, we leveraged a panel of four murine B-cell hybridoma lines capable of secreting mAb with specificity for the full-length SARS-CoV-2 Spike protein. Unlike in the context of B-cell immune-monitoring efforts, in which the response is polyclonal and the B-cell repertoire involved spans a spectrum of functional affinities for the antigen, the usage of B-cell hybridomas as model ASC enabled direct comparison of protein-coating efficiency in the context of ImmunoSpot assays, since these lines secrete monoclonal antibody (mAb) with defined specificity and epitope recognition for the Spike protein. As shown in Figure 3A, each of these B-cell hybridoma lines yielded clearly defined FluoroSpots when the assay well was coated with antibodies specific for the Igκ light chain; that is, when secreted immunoglobulin was detected irrespective of its antigen specificity. Counterstaining with IgG subclass-specific detection antibodies established that all four hybridomas secreted an IgG1, κ mAb. Next, we evaluated the capacity of these B-cell hybridoma lines to generate antigen-specific FluoroSpots in wells in which the Spike protein was directly captured at 25 or 10 µg/mL, or in wells coated at 10 µg/mL using the affinity capture approach. All four B-cell hybridoma lines were capable of antigen-specific FluoroSpot formation in wells coated directly with 25 µg/mL of the Spike protein. However, three of four B-cell hybridoma lines exhibited a significant reduction in spot-forming units (SFU) when seeded into wells coated directly with 10 µg/mL of Spike protein (Figure 3B). Importantly, the affinity capture of Spike protein yielded a significant improvement in the abundance of detectable FluoroSpots for the same three B-cell hybridoma lines. Furthermore, the FluoroSpot morphology was greatly enhanced for each of these model ASC when Spike protein was affinity captured (Figure 3C). Collectively, these data suggest that each of the 6XHis-tagged recombinant Spike antigen probes evaluated for utility in B-cell ImmunoSpot assays were more efficiently coated using an affinity capture approach compared to direct capture of these antigens to the assay membrane itself.

### 3.2. Affinity Tag Capture of EBV EBNA1 Protein Improves Detection of Antigen-Specific Memory B Cells

Encouraged by the improvements in detection of SARS-CoV-2 Spike-specific IgG^+^ ASC afforded through the affinity capture of 6XHis-tagged antigens, we next sought to establish B-cell ImmunoSpot assays for other viral antigens that could serve as specificity controls while monitoring for B-cell memory against SARS-CoV-2. As one such candidate virus, we attempted to detect pre-existing human memory B-cell reactivity against Epstein–Barr virus (EBV). Successful detection of memory B-cell reactivity against EBV is relevant in the context of immune monitoring since this herpesvirus causes infectious mononucleosis and is also linked to several hematopoietic malignancies [77,78]. Moreover, EBV can persist in a latent state within the memory B-cell compartment for the lifetime of a host [79]. Of particular interest, the Epstein–Barr nuclear antigen 1 (EBNA1) protein possesses multiple independent roles in viral latency (reviewed by [80]) and serum antibody reactivity to EBNA1 predominates during latent EBV infection [81].

Consistent with our observations using recombinant S1 protein, IgG^+^ FluoroSpots could be detected in wells in which EBNA1 protein was directly captured on the assay membrane (Figure 4A). However, in comparison, a greater number of EBNA1-specific IgG^+^ FluoroSpots were detected in wells coated using an affinity capture approach. Furthermore, the observed FluoroSpots in the affinity-captured antigen-coated wells possessed a greater fluorescence intensity and were larger in size compared to FluoroSpots detected in wells in which the EBNA1 protein was directly captured (Figure 4B). Therefore, affinity capture of EBNA1 protein improved detection of antigen-specific human ASC by all three criteria: spot number, spot intensity and spot size definition.

### 3.3. Affinity Tag Capture of HCMV gH Protein Improves Detection of Antigen-Specific Memory B Cells

As a second control antigen system for monitoring SARS-CoV-2-specific B cells, we also evaluated the efficiency of detecting memory B-cell reactivity against human cytomegalovirus (HCMV). In the context of immune monitoring of underlying memory B-cell reactivity against HCMV, we utilized a recombinant gH pentamer complex as our coating antigen since it is a known target of neutralizing antibodies [82]. As shown in Figure 5A, direct capture of the gH pentamer complex protein at 10 µg/mL failed to yield well-defined IgG^+^ FluoroSpots. Instead, the observed FluoroSpots were weakly fluorescent and diffuse, and barely countable. In support of the improvement in antigen coating afforded through affinity capture of the 6XHis-tagged gH pentamer complex, both an increase in the number of IgG^+^ FluoroSpots detected and enhancement of their fluorescence intensity was observed. Nevertheless, despite superior detection of gH-specific IgG^+^ ASC using the affinity capture coating approach, these FluoroSpots were still at the threshold of reliable detection. Therefore, we evaluated the hypothesis that increasing the quantity of both the anti-6XHis capture mAb and gH pentamer complex used for affinity capture coating would further improve the detection of gH-specific IgG^+^ ASC and enhance the intensity of the resulting FluoroSpots. As shown in Appendix A, and supporting our hypothesis, a significant increase in the number of gH-specific IgG^+^ ASC was detected in wells coated with 20 μg/mL of gH complex using an affinity capture approach. Furthermore, the FluoroSpots in these wells exhibited an improved morphology on the basis of fluorescence intensity and size. Collectively, these data highlight the improvement in assay sensitivity afforded through affinity capture of recombinant proteins via their encoded 6XHis affinity tag.

### 3.4. Affinity Tag Capture of Influenza Hemagglutinin Improves Antigen-Specific FluoroSpot Assays

As additional controls for monitoring SARS-CoV-2-specific B-cell memory, we also sought to evaluate the efficiency of detecting memory B-cell reactivity against recombinant hemagglutinin (rHA) proteins representing prototype H1N1 (A/California/04/2009, CA/09) and H3N2 (A/Texas/50/2012) influenza vaccine strains. Direct capture of CA/09 rHA protein at 10 µg/mL yielded well-defined IgG^+^ FluoroSpots for each of the four donors evaluated (Figure 6A). Additionally, affinity capture of CA/09 rHA did not further increase the number of detectable antigen-specific IgG^+^ FluoroSpots. However, affinity capture of CA/09 rHA did increase the fluorescence intensity and size of the observed FluoroSpots (Figure 6B). In contrast, affinity capture of TX/12 rHA protein afforded a clear improvement in the number of antigen-specific IgG^+^ ASC detected (Appendix A). Moreover, both the intensity and size of the observed antigen-specific IgG^+^ FluoroSpots were augmented in wells coated with TX/12 rHA through affinity capture.

To reconcile the discrepancy between the improvement in antigen coating observed in the context of TX/12 rHA with the observation that affinity capture of CA/09 rHA afforded only a modest benefit, we next evaluated the formation of CA/09 rHA-specific FluoroSpots using murine B-cell hybridomas as ASC. As noted previously, usage of B-cell hybridomas as ASC is an ideal approach to directly evaluate any improvements in protein-coating efficiency in the context of ImmunoSpot assays since these cells secrete monoclonal antibody (mAb) with defined specificity and epitope recognition for the CA/09 rHA protein. Moreover, the affinity of the secreted mAb is fixed and enables assessment of B-cell hybridoma lines known to secrete mAb with varying functional affinities. As shown in Figure 7A, each of the four selected murine B-cell hybridoma lines yielded well-defined FluoroSpots when their secreted antibody was captured irrespective of antigen specificity, and each was confirmed to secrete IgG1, κ. Three of these B-cell hybridoma lines were capable of generating well-defined CA/09 rHA-specific FluoroSpots when the protein was directly captured at 25 μg/mL. The fourth B-cell hybridoma line (4-G10.3) yielded only faint FluoroSpots in these directly coated wells, indicating a failure of the secreted mAb to be retained in close proximity to the secreting cell as a consequence of poor binding. Of the three lines that yielded definitive FluoroSpot formation in wells coated with 25 μg/mL of CA/09 rHA, two demonstrated a significant reduction in SFU in wells in which CA/09 rHA was directly captured at 10 μg/mL (Figure 7B). Importantly, affinity capture of CA/09 rHA at 10 μg/mL significantly improved FluoroSpot formation for both of these lines, and greatly enhanced FluoroSpot morphology (Figure 7C). Collectively, the data generated using the CA/09 rHA and TX/12 rHA proteins further support the notion that affinity capture of 6XHis-tagged proteins offers a robust strategy for achieving optimal antigen coating and reliable detection of antigen-specific ASC using the ImmunoSpot platform, and permits detection of ASC that produce antibodies of reduced functional affinity for an antigen of interest.

## 4. Discussion

Using an affinity capture coating approach leveraging the genetically encoded hexahistidine (6XHis) affinity tag commonly expressed by recombinant proteins, we introduce a universal platform for antigen absorption that improved the sensitivity of B-cell ImmunoSpot assays. Using murine B-cell hybridomas and in vitro differentiated human memory B cells as model ASC, we demonstrate the utility of the affinity capture approach across a variety of recombinant proteins: seasonal influenza (H1N1 and H3N2), latent herpesviruses (EBV and HCMV) and most notably the pandemic SARS-CoV-2 virus. Beyond reducing the concentration required to achieve efficient protein coating, as evidenced by increased antigen-specific SFU, the affinity capture approach also enhanced assay sensitivity based on augmented FluoroSpot intensity and size. Collectively, these data highlight the suitability of the FluoroSpot assay platform for high-throughput and systematic assessment of underlying memory B-cell reactivity, or in vivo differentiated ASC such as plasmablasts or LLPC, against diverse panels of recombinantly expressed protein antigens. Our data also show that affinity antigen coating was critical for developing high-resolution B-cell ImmunoSpot assays for SARS-CoV-2, including control antigens, and suggest that this approach may facilitate B-cell immune monitoring in general.

Development of antigen-specific B-cell memory following SARS-CoV-2 infection has already been evaluated using flow cytometry and fluorescently-conjugated probes [24,32,83], or following in vitro differentiation of donor PBMC and assessment of ASC reactivity in ELISPOT or FluoroSpot assays [84,85,86,87,88]. Our data demonstrating IgG^+^ ASC reactivity against the RBD and S1 coating antigens provides further evidence supporting the development of immunological memory. Using the affinity capture approach described in this communication, we readily detected RBD-specific ASC in all four donors tested (Figure 1). In contrast, ~25% of recovered COVID-19 donors evaluated using a nontraditional ELISPOT detection system encompassing an inverted assay approach and an HRP-conjugated RBD probe failed to yield detectable ASC reactivity [86]. The reliance on an inverted assay approach for detection of RBD-specific ASC is in agreement with our observation that direct, non-specific coating of SARS-CoV-2 RBD to the assay membrane is not an effective strategy for measuring ASC reactivity against this target. Importantly, whereas this inverted assay approach is restricted to assessment of IgG^+^ ASC [86,87], our affinity capture approach enables parallel assessment of multiple Ig classes or IgG subclasses when combined with multi-color FluoroSpot [37,58].

The Ig class usage of SARS-CoV-2-specific memory B-cell populations has also been evaluated using an antigen probe-based flow cytometric approach [24]. In addition to enabling identification of multiple antigen-specific populations in parallel, detailed information on the surface phenotype of such cells can also be acquired using this methodology. Nevertheless, development and implementation of such multi-parametric panels is technically complex and therefore this approach is unlikely to be suitable for performing large-scale immune-monitoring efforts. Furthermore, while the specificity of antigen probe binding is supported through the lack of appreciable staining observed using samples collected prior to the COVID-19 pandemic [24,32,33], a dedicated comparison between the lower limit of antigen-specific memory B-cell detection and their relative frequencies using a flow cytometric or an ImmunoSpot approach has not been reported to our knowledge. Furthermore, while such efforts are beyond the scope of this communication, the resulting data would shed invaluable insight into which of these complementary techniques is most appropriate for reliable detection of antigen-specific memory B-cell reactivity.

Exploitation of the 6XHis affinity tag for capture and subsequent purification of recombinant proteins is widely used [89]. Due to its small size, polyhistidine affinity tags are easily incorporated into either the N- or C-terminus of expression vectors, or the nucleotide sequence of the recombinant protein itself using polymerase chain reaction, and rarely does inclusion of this affinity tag affect protein function or structural integrity. Moreover, there are a variety of commercially available cloning vectors for the generation and expression of His-tagged recombinant proteins in different expression systems. The 6XHis affinity tag is only one of several previously described for subsequent purification of recombinant proteins [90]. Specifically, alternative affinity tags commonly used for purification of recombinant proteins include FLAG [91], HA [92], c-Myc [93] and Strep II [94]. In combination with the respective companion anti-tag mAb, or using a modified Strep-Tactin protein that binds the Strep II tag with high-affinity, these affinity tags may also lend themselves to similar usage in the context of antigen-specific ImmunoSpot assays. In this regard, further investigation into the applicability of alternative anti-tag capture systems for improved antigen absorption in the ImmunoSpot platform are merited, but are beyond the scope of this communication.

Initially commercialized for filtration applications, the porosity and high antibody-binding properties of polyvinylidene fluoride (PVDF) membranes render them ideally-suited for usage in ImmunoSpot applications [65]. According to general knowledge, proteins are absorbed on PVDF membranes through hydrophobic and dipole interactions. However, it stands to reason that usage of PVDF membranes may not be optimal for all B-cell ImmunoSpot applications due to the unique biochemistry and diversity of macromolecules that constitute the antigenic universe. In this context, the affinity capture approach described in this communication circumvents the variability introduced through absorption of proteins on the basis of their unique conformation and amino acid sequence composition. Instead, a diversity of recombinant proteins bearing the 6XHis or an alternative affinity tag may be efficiently coated in a relatively straightforward and streamlined fashion through preconditioning the assay well first with the corresponding high-affinity capture reagent, followed by subsequent acquisition of soluble protein.

Conceptually, the affinity capture approach may improve assay sensitivity and reduce the concentration required to detect antigen-specific ASC by better preserving the tertiary (or quaternary) structure of coating antigens through directing their interaction with the capture surface to a defined footprint. In this way, the chance that absorption of antigen directly to the PVDF membrane could result in protein distortion and subsequent destruction of relevant epitopes would be reduced. Moreover, the affinity-tagged protein is likely to project away from the membrane and this could avoid steric hinderance and enhance accessibility of the secreted antibody for certain epitopes. In this context, oriented immobilization and capture of proteins via an affinity tag is not a novel idea and has broader applications beyond improving antigen coating in the context of ImmunoSpot (reviewed by [95]).

Another benefit of the affinity capture approach may be the ability to achieve a higher local density of antigen in close proximity to each other relative to that achieved when antigen is applied directly to the assay membrane. Since retention of the secreted antibody in close proximity to the source cell is essential for detection of antigen-specific ImmunoSpots, especially for ASC with reduced functional affinity, improvements in antigen spatial packing are in line with our observations of improved assay sensitivity. In support of this hypothesis, we observed distinct increases in the intensity of antigen-specific FluoroSpot in several donors and murine B-cell hybridoma lines.

With the exception of the CA/09 rHA coating antigen, we present evidence for improved sensitivity in the detection of underlying human memory B-cell reactivity against several viral pathogens using an affinity capture approach. This observation could suggest that the CA/09 rHA protein, unlike the other recombinant proteins used as coating antigens for B cell FluoroSpot assays, was efficiently captured directly on the assay membrane. However, in light of the increased SFU and FluoroSpot intensity observed using CA/09 rHA-specific murine B-cell hybridomas as model ASC (Figure 7), we consider this explanation unlikely. Alternatively, this result could merely be due to the limited number of donors that were evaluated in parallel (*n* = 4) for FluoroSpot formation against the CA/09 rHA. Instead, we speculate that CA/09 rHA-specific ASC in these human donors were endowed with increased affinity. Supporting the hypothesis that CA/09 rHA-specific ASC possess increased affinity due to previous affinity maturation; the human population is routinely exposed to H1N1 viruses through natural infection or prophylactic vaccination [96]. Furthermore, the H1N1 component in the annual seasonal influenza vaccine was antigenically matched or was quite similar to the CA/09 rHA for the past decade. To this end, a more detailed assessment of CA/09 rHA-specific ASC and their resulting FluoroSpot morphologies using a larger donor cohort pool would resolve this open question, but was beyond the scope of this communication.

As an additional component of our antigen-specific FluoroSpot analysis pipeline, we leveraged the ImmunoSpot^®^ counting software’s capacity to export flow cytometry standard (FCS) files and subsequently segregated individual FluoroSpots on the basis of their intensity and size. Visualizing the FluoroSpot data in a bivariate plot representing these parameters, and overlaying the counted FluoroSpot events occurring in wells in which the antigen was directly or affinity captured, enabled a clear visualization of the qualitative changes afforded through affinity capture coating of the antigen. Through extension, and based on mathematical modeling, there should be a relationship between the size and intensity (collectively termed morphology) of antigen-specific FluoroSpots derived from ASC with differential affinity [60]. Improvements in antigen absorption in this context are thus invaluable since they conceptually extend the window of observation through enabling detection of FluoroSpots originating from ASC with reduced functional affinity and would offer insights into whether recall responses exhibit evidence of affinity maturation [97]. Nevertheless, due to the polyclonal nature and diversity of epitopes recognized by individual antigen-specific human ASC, experimental efforts intending to confirm and extend the association between antibody functional affinity and FluoroSpot morphology are best resolved using model ASC that span a defined range of affinity and recognize a common epitope (Becza et al., manuscript in preparation). Furthermore, while an association between mAb functional affinity and FluoroSpot formation/morphology was not readily apparent using the limited number of CA/09 rHA- and SARS-CoV-2 Spike-specific B-cell hybridoma lines detailed in this communication, assessment of a larger collection of model ASC will be needed to lend support to this hypothesis.

The improvements in antigen coating described in this communication also serve to elevate the utility of the FluoroSpot platform itself, since multiple Ig classes or IgG subclasses can be evaluated in parallel through multiplexing of detection systems [37,58]. Moreover, beyond permitting identification of Ig class or IgG subclass usage of individual antigen-specific ASC, the ImmunoSpot assay platform also serves as a key methodology to bridge assessments of acute B-cell responses. ImmunoSpot assays permit the detection of plasmablasts directly ex vivo through their spontaneous antibody secretion. In contrast, the memory B-cell component, consisting of resting lymphocytes, can be detected via its ASC activity only following in vitro differentiation. Moreover, ongoing optimizations in the cryopreservation of resting B cells and B-cell blasts [67,98], along with the high-throughput capability of the ImmunoSpot approach, also enable simultaneous assessments of these respective compartments and detailed comparisons, such as Ig class usage and relative abundance amongst the total ASC response. In this context, it is important to note that acutely following infection or vaccination, the majority of circulating plasmablasts will be endowed with high-affinity, antigen-specific BCR since they were successfully recruited into the T-cell-dependent immune response [4,99]. In stark contrast, ASC with reactivity against an antigen of interest may comprise a very small frequency of the total ASC compartment and are likely to possess a wider spectrum of affinity following in vitro polyclonal stimulation, since these methods promote B-cell differentiation independent of BCR-encoded specificity. Therefore, it is imperative to introduce the appropriate negative controls into assays aiming at revealing underlying memory B-cell reactivity, especially those bearing an unswitched IgM^+^ BCR, following in vitro differentiation of donor PBMC [88]. Furthermore, a B-cell stimulation protocol such as anti-CD40 + IL-4 + IL-21 that mimics physiologic B-cell stimulation through helper T cells, rather than reliance on mitogens or TLR agonists, enables selective activation of memory B cells [37] and may offer a superior strategy for identifying antigen-specific IgM^+^ memory B cells following resolution of viral infections such as COVID-19.

The presence of LLPC secreting SARS-CoV-2 Spike-reactive IgA or IgG was recently demonstrated using a double-color enzymatic ImmunoSpot approach [15]. In agreement with the role of LLPC, the abundance of serum antibody reactivity against the Spike antigen correlated with the frequency of antigen-specific plasma cells detected in bone marrow aspirates. However, in comparison to the high-quality ELISPOTs generated when the secreted IgA or IgG was captured irrespective of its antigen specificity, the antigen-specific spots depicted in this report were faint and diffuse. Based on the data presented in this communication, we speculate that these assays were performed using suboptimal antigen-coating conditions. Consequently, not only were these assay results likely difficult to enumerate, it is plausible that the magnitude of the Spike-reactive LLPC compartment was underestimated, since plasma cells secreting lower-affinity specificities may not have yielded a detectable secretory footprint.

In closing, the affinity capture approach introduced in this communication opens the door to systematic assessment of underlying memory B-cell reactivity against a wide array of antigens, regardless of their conformation and amino acid composition using the ImmunoSpot platform. Such assessment might be essential for understanding immunity to SARS-CoV-2. Our ongoing studies reveal that low serum antibody reactivity to SARS-CoV-2 poorly reflects on the abundance of SARS-CoV-2 antigen-specific memory B cells in subjects who recovered from infection, and that such memory B cells can be regularly detected in recovered subjects even if the serum antibody levels were too low to confirm prior exposure (Wolf et al., manuscript in preparation). As recent reports indicate that antibody titers against the SARS-CoV-2 Spike protein may decline over the course of months [22,23,24,100], we suggest that the relevant assessment of B-cell immunity in the context of the ongoing global pandemic exists at the level of memory B cells that can convey immune protection by engaging in a rapid secondary antibody response. We further contend that ImmunoSpot offers a suitable platform for the detailed study of SARS-CoV-2 Spike-reactive memory B cells and suggest its overall utility for identifying protective immune signatures acquired either through natural infection or following prophylactic vaccination.

## Figures and Tables

**Figure 1 cells-10-01843-f001:**
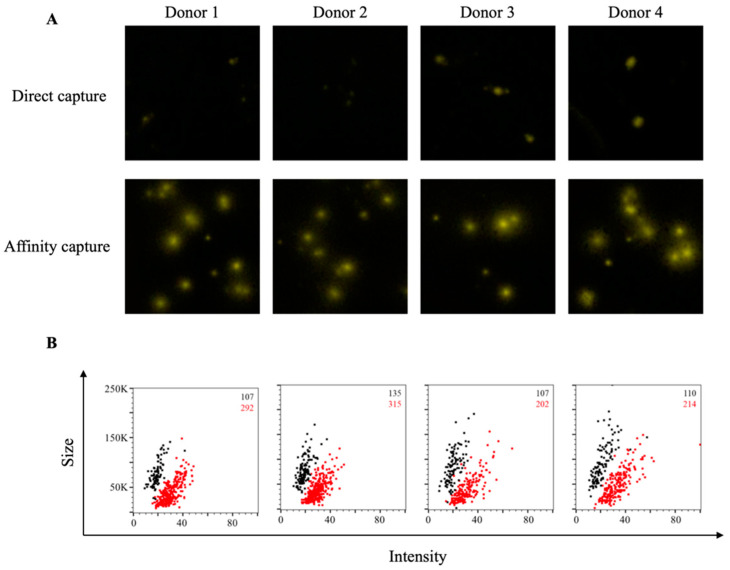
Affinity tag capture improves detection of SARS-CoV-2 RBD-reactive ASC. PBMC from four PCR-confirmed COVID-19 donors were stimulated in vitro (detailed in Materials and Methods) and evaluated for antibody-secreting (ASC) reactivity against the receptor-binding domain (RBD) fragment of the SARS-CoV-2 Spike protein. (**A**) Representative well images depicting antigen-specific IgG^+^ ASC in wells coated directly with 10 μg/mL of RBD protein or through affinity capture using the genetically encoded hexahistidine (6XHis) tag. Magnification and contrast enhancements were uniformly performed on all images to aid their visualization in publication. (**B**) RBD-specific FluoroSpots were merged into flow cytometry standard (FCS) files (detailed in Materials and Methods) and visualized as bivariate plots measuring spot intensity (x-axis) and spot size (y-axis). FluoroSpots originating from assay wells in which RBD protein was directly captured on the membrane (black dots) or through affinity capture (red) are shown as overlays. The combined number of FluoroSpots (spot-forming units, SFU) detected in replicate wells for each of the respective donors is indicated in the inset using the same red/black color code.

**Figure 2 cells-10-01843-f002:**
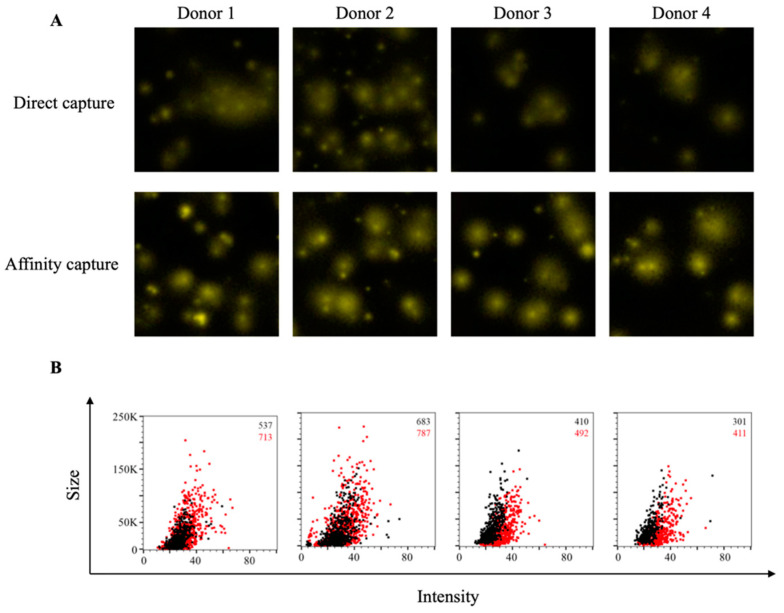
Affinity tag capture improves detection of SARS-CoV-2 S1-reactive ASC. PBMC from four PCR-confirmed COVID-19 donors were stimulated in vitro and evaluated for antibody-secreting (ASC) reactivity against the S1 subunit of the SARS-CoV-2 Spike protein. (**A**) Representative well images depicting antigen-specific IgG^+^ ASC in wells coated directly with 10 μg/mL of S1 protein or through affinity capture using the genetically encoded hexahistidine (6XHis) tag. Magnification and contrast enhancements were uniformly performed on all images to aid their visualization in publication. (**B**) S1-specific FluoroSpots were merged into FCS files and visualized as bivariate plots. FluoroSpots originating from assay wells in which S1 protein was directly captured on the membrane (black dots) or through affinity capture (red) are shown as overlays. The combined number of FluoroSpots detected in replicate wells for each of the respective donors is indicated in the inset using the same red/black color code.

**Figure 3 cells-10-01843-f003:**
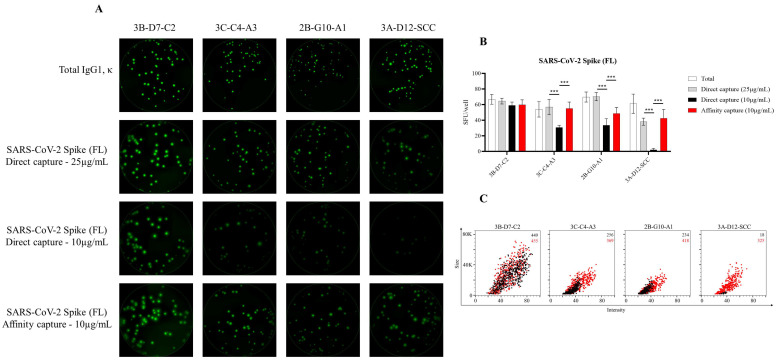
Detection of SARS-CoV-2 Spike-specific FluoroSpots is influenced by protein-coating efficiency. Murine B-cell hybridomas (~100 cells/well) were evaluated for total or antigen-specific FluoroSpot formation (detailed in Materials and Methods). (**A**) Representative well images of murine B-cell hybridomas secreting monoclonal antibody (mAb) (IgG1, κ) with specificity for the full-length (FL) SARS-CoV-2 Spike protein. Contrast enhancements were uniformly performed on all images to aid their visualization in publication. (**B**) Total or Spike-specific SFU/well (mean ±SD) for each B-cell hybridoma line. Significant differences in SFU/well were determined using an analysis of variation (ANOVA) with Sidak’s post hoc test. *** *p* < 0.001. (**C**) Spike-specific FluoroSpots were merged into FCS files and visualized as bivariate plots. FluoroSpots originating from assay wells in which Spike protein was directly captured on the membrane (black dots) or through affinity capture (red) are shown as overlays. The combined number of FluoroSpots detected in replicate wells for each of the respective donors is indicated in the inset using the same red/black color code.

**Figure 4 cells-10-01843-f004:**
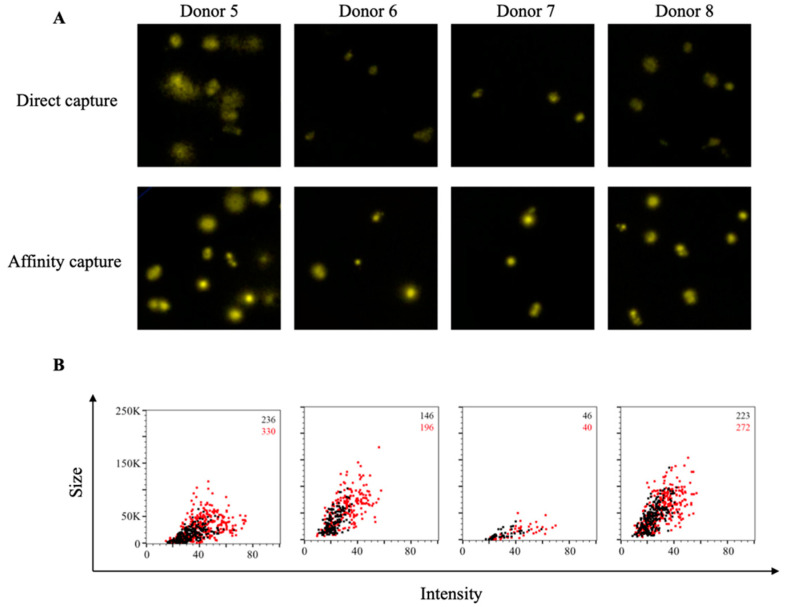
Improved sensitivity of EBNA1-specific ASC through affinity capture of coating antigen. PBMC were stimulated in vitro and evaluated for antibody-secreting (ASC) reactivity against the Epstein–Barr nuclear antigen 1 (EBNA1) protein from EBV. (**A**) Representative well images depicting antigen-specific IgG^+^ ASC in wells coated directly with 10 μg/mL of EBNA1 protein or through affinity capture using the genetically encoded hexahistidine (6XHis) tag. Magnification and contrast enhancements were uniformly performed on all images to aid their visualization in publication. (**B**) EBNA1-specific FluoroSpots were merged into FCS files and visualized as bivariate plots. FluoroSpots originating from assay wells in which EBNA1 protein was directly captured on the membrane (black dots) or through affinity capture (red) are shown as overlays. The combined number of FluoroSpots detected in replicate wells for each of the respective donors is indicated in the inset using the same red/black color code.

**Figure 5 cells-10-01843-f005:**
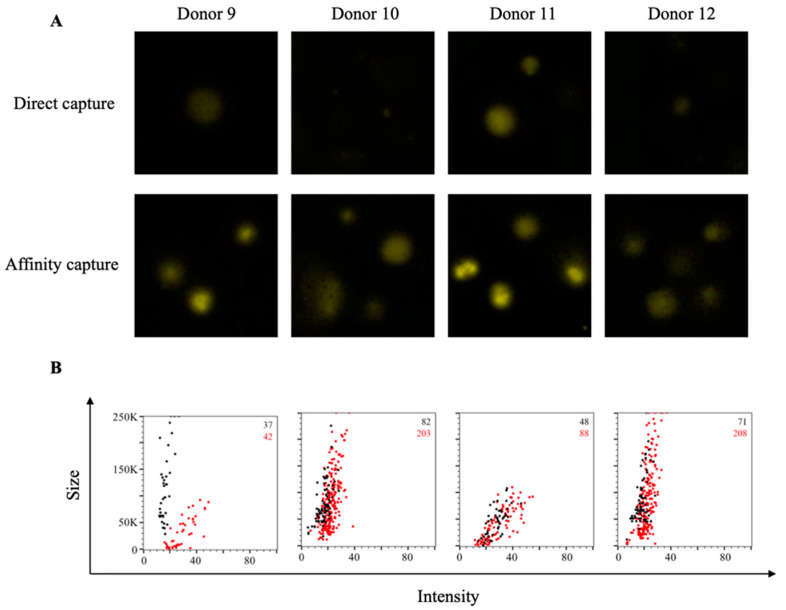
Improved sensitivity of gH-specific ASC through affinity capture of coating antigen. PBMC were stimulated in vitro and evaluated for antibody-secreting (ASC) reactivity against the gH pentamer complex protein from HCMV. (**A**) Representative wells images depicting antigen-specific IgG^+^ ASC in wells coated directly with 10 μg/mL of gH pentamer complex protein, or through affinity capture using the genetically encoded hexahistidine (6XHis) tag. Magnification and contrast enhancements were uniformly performed on all images to aid their visualization in publication. (**B**) gH-specific FluoroSpots were merged into FCS files and visualized as bivariate plots. FluoroSpots originating from assay wells in which gH pentamer complex was directly captured on the membrane (black dots) or through affinity capture (red) are shown as overlays. The combined number of FluoroSpots detected in replicate wells for each of the respective donors is indicated in the inset using the same red/black color code.

**Figure 6 cells-10-01843-f006:**
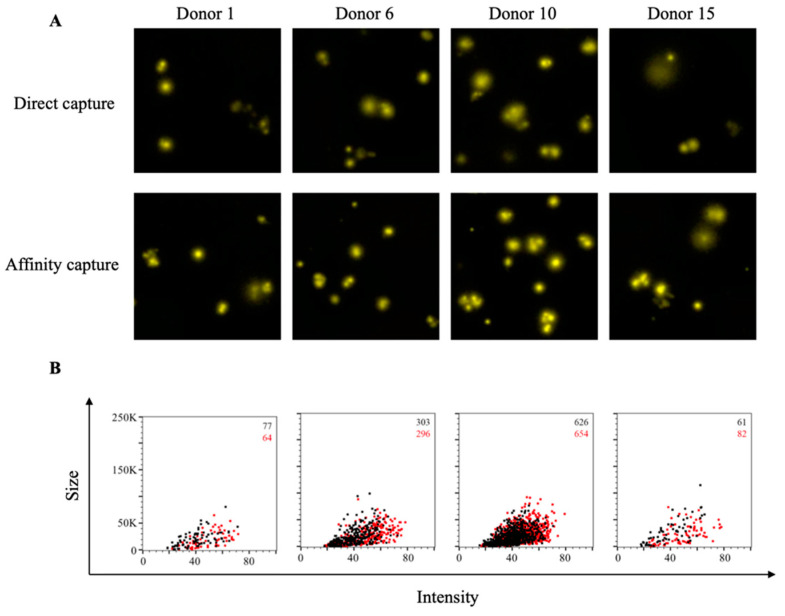
Improved sensitivity of CA/09 rHA-specific ASC through affinity capture of coating antigen. PBMC were stimulated in vitro and evaluated for antibody-secreting (ASC) reactivity against recombinant hemagglutinin protein representing the A/California/04/2009 (CA/09) H1N1 vaccine strain. (**A**) Representative well images depicting antigen-specific IgG^+^ ASC in wells coated directly with 10 μg/mL of CA/09 rHA or through affinity capture using the genetically encoded hexahistidine (6XHis) tag. Magnification and contrast enhancements were uniformly performed on all images to aid their visualization in publication. (**B**) CA/09 rHA-specific FluoroSpots were merged into FCS files and visualized as bivariate plots. FluoroSpots originating from assay wells in which CA/09 rHA was directly captured on the membrane (black dots) or through affinity capture (red) are shown as overlays. The combined number of FluoroSpots detected in replicate wells for each of the respective donors is indicated in the inset using the same red/black color code.

**Figure 7 cells-10-01843-f007:**
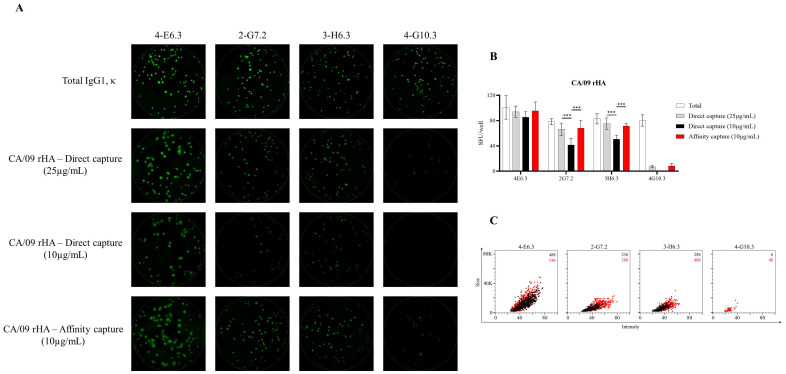
Detection of CA/09 rHA-specific FluoroSpots is influenced by protein-coating efficiency. Murine B-cell hybridomas (~100 cells/well) were evaluated for total or antigen-specific FluoroSpot formation. (**A**) Representative well images of murine B-cell hybridomas secreting monoclonal antibody (mAb) (IgG1, κ) with specificity for the recombinant hemagglutinin protein representing the A/California/04/2009 (CA/09) H1N1 vaccine strain. Contrast enhancements were uniformly performed on all images to aid their visualization in publication. (**B**) Total or CA/09 rHA-specific SFU/well (mean ±SD) for each B-cell hybridoma line. Significant differences in SFU/well were determined using an analysis of variation (ANOVA) with Sidak’s post hoc test. *** *p* < 0.001. (**C**) CA/09 rHA-specific FluoroSpots were merged into FCS files and visualized as bivariate plots. FluoroSpots originating from assay wells in which CA/09 rHA was directly captured on the membrane (black dots) or through affinity capture (red) are shown as overlays. The combined number of FluoroSpots detected in replicate wells for each of the respective donors is indicated in the inset using the same red/black color code.

## Data Availability

The data generated in this study will be made available by the authors, without undue reservation, to any qualified researcher.

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
