# Peer review of "Affinity Tag Coating Enables Reliable Detection of Antigen-Specific B Cells in Immunospot Assays"

_cells, 2021, doi:10.3390/cells10081843_

Round 1

Author Response

We thank the Reviewers for their favorable review of our manuscript entitled “Affinity tag coating enables reliable detection of antigen-specific B cells in ImmunoSpot assays”. In the following, please find our point-by-point response to the Reviewer comments. Additionally, we specify the corresponding line numbers in which we introduced/modified text and highlight these changes in green in the revised submission.

Reviewer #1:

We thank Reviewer #1 for his/her overall favorable assessment of our manuscript. The summary of our major findings was accurate and his/her final comment “experimental results are adequate and convincing, and the manuscript is well-written” is greatly appreciated.

Reviewer #1 did specify a few “minor” comments in regards to our manuscript, and we provide responses and highlight modifications in the revised manuscript that were introduced in response to Reviewer #1’s feedback.

Point #1: “The authors claim that this assay allows an unambiguous detection of memory B cells. However, there is no evidence showing that memory B cells became ASCs, or the percentage of memory B cells became ASCs, using the method as described in Materials and Methods 2.2.

Response: We thank Reviewer #1 for this comment and the opportunity to offer further clarification in regards to our statement. Stimulation of PBMC with TLR7/8 agonist R848 + recombinant IL-2 (B-Poly-S™) is an established method for driving terminal differentiation of pre-existing memory B cells into ASC (PMID 19404981). To this extent

we cite existing literature that evaluated surface phenotype of B cells during the course of in vitro stimulation using T cell independent stimulation approaches (PMID 23585684 and PMID 31844520). These data, along with our own unpublished observations using R848 + rIL-2, indicate that B cells do not uniformly acquire a fully differentiated plasma cell (CD138+) phenotype after short-term (4-6 days) in vitro stimulation. Therefore, the Reviewer is correct, R848 +rIL2 stimulation is not a perfect way of reactivating memory B cells, but it is the best, according to the literature. Consequently, we routinely report the abundance of antigen-specific ASC following in vitro stimulation as the percentage of Total ASC activity to account for donor variability in the efficiency of polyclonal stimulation and terminal ASC differentiation.

Point #2: “the level of long-lived plasma cells (LLPC) of these samples are also unknown. Therefore, the authors are recommended to perform a profiling experiment of human PBMCs before and after stimulation.”  

Response: Reviewer #1 is correct, the level of LLPC in the samples evaluated in our manuscript were unknown. Unfortunately, LLPC reside predominantly in the bone marrow and our studies were limited to usage of PBMC. Therefore, assessment of LLPC in these subjects is not possible under our IRB constraints.

In response to Reviewer #1’s recommendation that we perform profiling experiments of human PBMC before and after stimulation to monitor differentiation, we agree that these data would be relevant in the context of tracking the efficiency B cell differentiation in vitro. The most telling experiments would be tracking antigen-specific B cells as identified by staining with labeled antigen, which is not a simple undertaking. We have such experiments in the working, but conclusive data are still months ahead. Therefore, we hope the Reviewer agrees that we present such data in a future manuscript, in particular because they do not affect the conclusion of this one.

Point #3: “In the abstract, the sentence “There are few techniques that are capable of detecting rare antigen-specific B cells while also providing information regarding their precursory frequency, class/sub-class usage and functional affinity” gives the wrong impression that the assay described in this manuscript can achieve these goals. However, there are no results showing the affinity tag capture technique can unveil the precursory frequency, class/sub-class usage and functional affinity. Therefore, a clarification is needed”

Response: We thank Reviewer #1 for this critique and opportunity to offer a clarification to improve our revised manuscript.  ELISPOT/FluoroSpot assays detect antibody secretion by individual ASC, and thus they are strictly quantitative, but the Reviewer is correct: precursor frequencies are only indirectly revealed. Therefore

we have modified the sentence in the abstract as follows;

“There are few techniques that are capable of detecting rare antigen-specific B cells while also providing information regarding their relative abundance, class/sub-class usage and functional affinity”

  • We have also modified a sentence in the Discussion, starting on Line 584

“Moreover, ongoing optimizations in cryopreservation of resting B cells and B cell blasts [66, 90], along with the high-throughput capability of the ImmunoSpot approach, also enable simultaneous assessments of these respective compartments and detailed comparisons; such as Ig class usage and relative abundance amongst the total ASC response.

  • Lastly, we introduced the following text starting on Line 573 of the revised manuscript.

“The improvements in antigen coating described in this communication also serve to elevate the utility of the FluoroSpot platform itself since multiple Ig classes or IgG subclasses can be evaluated in parallel through multiplexing of detection systems [37,58].

Reviewer 2 Report

The authors developed a modification of ELISpot assay for detection of antigen-specific B cells. The novelty of the approach arises in the use of the affinity capture of antigen in contrast to the non-specific random antigen absorption in the standard ELISpot assay. The attractive side of the manuscript is that the advances of proposed method has been demonstrated for several antigens, among which the coronavirus spike antigen attracts the most attention. The study is of interest and very relevant with the current pandemic. Comparison with direct non-specific absorption showed this approach to be more sensitive than standard ELISpot, correctly identifying more antigen-specific B cells. A good statistical analysis of the results obtained has been carried out.

I have a few minor comments about the manuscript:

1 The paper is purely methodical. The idea of oriented immobilization of antigen or antibodies is not new. Previously, this approach was used in ELISA, as well as in the production of immunoaffinity sorbents and biosensors (for review see for example Yingshuai Liu & Jie Yu Oriented immobilization of proteins on solid supports. Microchim Acta (2016) 183:1–19).

An alternative method for the determination of antigen-specific B memory cells is the binding of fluorescently labeled antigens (see for example Dan JM et al., Immunological memory to SARS-CoV-2 assessed for up to 8 months after infection. Science. 2021 Feb 5;371(6529):eabf4063. doi: 10.1126/science.abf4063). The scientific significance of the manuscript could be increased by comparing the ELISpot results with the detection of antigen-specific B cells according to binding of fluorescently labeled antigens. It would be interesting to know which method (affinity capture or direct non-specific antigen absorption) correlates better with flow cytometric antigen binding data. It would be ideal to compare these methods in parallel experiments or at least by comparison with literature data.

2 The authors point out that there are several ways to activate B memory cells in vitro (using different combinations of IL-21, CD40L, IL-4, and TLR agonists). In paper, B cells were stimulated with B-Poly-STM reagent (from CTL). To compare the presented results with the literature data, it would be desirable to indicate what components this reagent consists of.

3 The attractive side of the article is the application of the proposed method for assessing coronavirus-specific B cells in COVID-19 convalescent. The use of the ELISpot assay for these purposes is still quite rare. We can recommend that the authors refer to the few articles that have already been published on this issue.

Discussing long-lived plasma cells (LLPC), the authors refer to the work of Seow J, et al. Longitudinal observation and decline of neutralizing antibody responses in the three months following SARS-CoV-2 infection in humans. Nature Microbiology. 2020; 5 (12): 1598-607, in which the level of serum antibodies was measured. More relevant work is available (Turner JS, at al., SARS-CoV-2 infection induces long-lived bone marrow plasma cells in humans. Nature. 2021 May 24. doi: 10.1038/s41586-021-03647-4.), in which the LLPC was defined using ELISpot assay.

The manuscript has the following keywords: ELISpot; memory B cell; plasmablast; SARS-CoV-2. Using these keywords two relevant article are identified and which it will be helpful to cite:

Byazrova et al., Pattern of circulating SARS-CoV-2-specific antibody-secreting and memory B-cell generation in patients with acute COVID-19. Clin Transl Immunology. 2021 Jan 31;10(2):e1245. doi: 10.1002/cti2.1245.

Yao L at al., Persistence of Antibody and Cellular Immune Responses in COVID-19 patients over Nine Months after Infection. J Infect Dis. 2021 May 12:jiab255. doi: 10.1093/infdis/jiab255

Author Response

We thank Reviewer #2 for their evaluation of our manuscript and we are pleased to receive such positive feedback. His/Her comment “The attractive side of the manuscript is that the advances of proposed method has been demonstrated for several antigens, among which the coronavirus spike antigen attracts the most attention” is greatly appreciated.

Reviewer #2 did specify a few “minor” comments in regards to our manuscript, and we provide responses and highlight modifications in the revised manuscript that were introduced in response to Reviewer #2’s feedback.

Point #1: “The paper is purely methodical. The idea of oriented immobilization of antigen or antibodies is not new. Previously, this approach was used in ELISA, as well as in the production of immunoaffinity sorbents and biosensors (for review see for example Yingshuai Liu & Jie Yu Oriented immobilization of proteins on solid supports. Microchim Acta (2016) 183:1–19).”

Response: Reviewer #2 is correct; the concept of oriented immobilization of antigen or antibodies is not novel. However, based on our current knowledge, our manuscript serves as the first description of this technique for improving protein absorption in the context of antigen-specific B cell ImmunoSpot. We appreciate Reviewer #2’s suggestion of citing the review authored by Yingshuai Liu and Jie Yu (https://doi.org/10.1007/s00604-015-1623-4)

  • We have introduced the following sentence into the discussion of the revised manuscript (starting on Line 524) as follows;

“In this context, oriented immobilization and capture of proteins via an affinity tag is not a novel idea and has broader applications beyond improving antigen coating in the context of ImmunoSpot (reviewed by [95]).

Point #2: “An alternative method for the determination of antigen-specific B memory cells is the binding of fluorescently labeled antigens (see for example Dan JM et al., Immunological memory to SARS-CoV-2 assessed for up to 8 months after infection. Science. 2021 Feb 5;371(6529):eabf4063. doi: 10.1126/science.abf4063). The scientific significance of the manuscript could be increased by comparing the ELISpot results with the detection of antigen-specific B cells according to binding of fluorescently labeled antigens. It would be interesting to know which method (affinity capture or direct non-specific antigen absorption) correlates better with flow cytometric antigen binding data. It would be ideal to compare these methods in parallel experiments or at least by comparison with literature data.”

Response: We thank Reviewer #2 for this comment and referencing the alternative flow cytometry-based approach for identifying antigen-specific B cells using fluorescently-labeled probes. We are very familiar with this approach and have first-hand experience using this technique in the context of studying seasonal influenza vaccine-elicited B cell responses (PMID 31794433). Additionally, we cited several papers that utilized a flow cytometry probe-based approach for identification of antigen-specific B cells in our manuscript, including the publication of Dan et al. that was indicated by Reviewer #2.

We assume that the results of either approach will depend on the concentration of antigen used for detection of antigen-specific B cells. If a high concentration of labeled probe were used for flow cytometric detection, B cells with low affinity for the antigen may also be stained. Consequently, the frequency of antigen-specific B cells would be higher than after staining with a lower concentration of the same probe. For ELISPOT/FluoroSpot. We know that low affinity B cells will are detectable only if the antigen coating density is high. Reviewer #2 is correct in their assumption that assay conditions will influence the absolute number of antigen-specific B cells, but we expect that detailed analysis – primarily focusing on affinity- are required to shed light on such differences. While we have such affinity-oriented studies on the way, we are far from solidifying this conclusion, and we hope the Reviewer agrees that once such conclusions are reached, they merit a manuscript on their own.

Nevertheless, we did include additional text in the Discussion of our manuscript as follows (starting on Line 461);

“Development of antigen-specific B cell memory following SARS-CoV-2 infection has already been evaluated using flow cytometry and fluorescently-conjugated probes [24,32,83], or following in vitro differentiation of donor PBMC and assessment of ASC reactivity in ELISPOT or FluoroSpot assays [84-88]. Our data demonstrating IgG+ ASC reactivity against the RBD and S1 coating antigens provides further evidence supporting the development of immunological memory. Using the affinity capture approach described in this communication, we readily detected RBD-specific ASC in all four donors tested (Figure 1). In contrast, ~25% of recovered COVID-19 donors evaluated using a nontraditional ELISPOT detection system encompassing an inverted assay approach and an HRP-conjugated RBD probe failed to yield detectable ASC reactivity [86]. The reliance on an inverted assay approach for detection of RBD-specific ASC is in agreement with our observation that direct, non-specific coating of SARS-CoV-2 RBD to the assay membrane is not an effective strategy for measuring ASC reactivity against this target. Importantly, whereas this inverted assay approach is restricted to assessment of IgG+ ASC (REF), our affinity capture approach enables parallel assessment of multiple Ig class or IgG subclass when combined with multi-color FluoroSpot [37,58].

The Ig class usage of SARS-CoV-2-specific memory B cell populations has also been evaluated using an antigen probe-based flow cytometric approach [24]. In addition to enabling identification of multiple antigen-specific populations in parallel, detailed information on the surface phenotype of such cells can also be acquired using this methodology. Nevertheless, development and implementation of such multi-parametric panels is technically complex and therefore this approach is unlikely to be suitable for performing large-scale immune monitoring efforts. Furthermore, while the specificity of antigen probe binding is supported through the lack of appreciable staining observed using samples collected prior to the COVID-19 pandemic [24,32,33], a dedicated comparison between the lower limit of antigen-specific memory B cell detection and their relative frequencies using a flow cytometric or an ImmunoSpot approach has not been reported to our knowledge. Furthermore, while such efforts are beyond the scope of this communication, the resulting data would shed invaluable insight into which of these complementary techniques is most appropriate for reliable detection of antigen-specific memory B cell reactivity.

Point #3: “The authors point out that there are several ways to activate B memory cells in vitro (using different combinations of IL-21, CD40L, IL-4, and TLR agonists). In paper, B cells were stimulated with B-Poly-S™ reagent (from CTL). To compare the presented results with the literature data, it would be desirable to indicate what components this reagent consists of.”

Response: We thank Reviewer #2 for bringing this to our attention. We modified the text in the Materials and Methods Section 2.2 (Line 196) to specify that B-Poly-S™ includes TLR7/8 agonist R848 + recombinant human IL-2. Of note, this approach is considered the most potent modality for polyclonal B cell stimulation (PMID 19404981).

Point #4: “The attractive side of the article is the application of the proposed method for assessing coronavirus-specific B cells in COVID-19 convalescent. The use of the ELISpot assay for these purposes is still quite rare. We can recommend that the authors refer to the few articles that have already been published on this issue.”

“Discussing long-lived plasma cells (LLPC), the authors refer to the work of Seow J, et al. Longitudinal observation and decline of neutralizing antibody responses in the three months following SARS-CoV-2 infection in humans. Nature Microbiology. 2020; 5 (12): 1598-607, in which the level of serum antibodies was measured. More relevant work is available (Turner JS, at al., SARS-CoV-2 infection induces long-lived bone marrow plasma cells in humans. Nature. 2021 May 24. doi: 10.1038/s41586-021-03647-4.), in which the LLPC was defined using ELISpot assay.”

Response: We thank Reviewer #2 for bringing this recent paper to our attention. We have introduced this reference into our manuscript to support the utility of the ImmunoSpot approach for detection of antigen-specific ASC. Additionally, we introduced the following text into our revised manuscript (starting on Line 598)

“The presence of LLPC secreting SARS-CoV-2 Spike-reactive IgA or IgG was recently demonstrated using a double-color enzymatic ImmunoSpot approach [15]. In agreement with the role of LLPC, the abundance of serum antibody reactivity against the Spike antigen correlated with the frequency of antigen-specific plasma cells detected in bone marrow aspirates. However, in comparison to the high-quality ELISPOTs generated when the secreted IgA or IgG was captured irrespective of its antigen-specificity, the antigen-specific spots depicted in this report were faint and diffuse. Based on the data presented in this communication, we speculate that these assays were performed using suboptimal antigen coating conditions. Consequently, not only were these assay results likely difficult to enumerate, it is plausible that the magnitude of the Spike-reactive LLPC compartment was underestimated since plasma cells secreting lower-affinity specificities may not have yielded a detectable secretory foot-print.”

Point #5: The manuscript has the following keywords: ELISpot; memory B cell; plasmablast; SARS-CoV-2. Using these keywords two relevant article are identified and which it will be helpful to cite:

Byazrova et al., Pattern of circulating SARS-CoV-2-specific antibody-secreting and memory B-cell generation in patients with acute COVID-19. Clin Transl Immunology. 2021 Jan 31;10(2):e1245. doi: 10.1002/cti2.1245.

Yao L at al., Persistence of Antibody and Cellular Immune Responses in COVID-19 patients over Nine Months after Infection. J Infect Dis. 2021 May 12:jiab255. doi: 10.1093/infdis/jiab255”

Response:  We thank Reviewer #2 for bringing both of these references to our attention, and they are both now cited in our revised manuscript.

Round 2

Reviewer 1 Report

The revision has addressed the comments reasonably. The reviewer has no further comments and thus suggest an acceptance.

Reviewer 2 Report

Nevertheless, I can report that the quality of the submitted paper was high and my comments were minor.
I am sure that the authors took into account my comments and the paper can be accepted for publication.